# Pro-Health Behaviours and Depressive Symptoms as Well as Satisfaction with and Quality of Life Among Women with Hashimoto’s Disease

**DOI:** 10.3390/ejihpe15060097

**Published:** 2025-06-02

**Authors:** Maria Gacek, Agnieszka Wojtowicz, Jolanta Kędzior

**Affiliations:** 1Department of Sports Medicine and Human Nutrition, University of Physical Culture in Krakow, Jana Pawła II 78, 31-571 Krakow, Poland; 2Department of Psychology, University of Physical Culture in Krakow, Jana Pawła II 78, 31-571 Krakow, Poland; agnieszka.wojtowicz@awf.krakow.pl; 3College of Physical Education and Sport, University of Bielsko-Biała, Willowa 2, 43-309 Bielsko-Biała, Poland

**Keywords:** lifestyle, hypothyroidism, mental health, quality of life

## Abstract

Background: Lifestyle is one of the important factors determining health and quality of life. The aim of the study was to analyse relationships between pro-health behaviours, depression and quality of life among women with Hashimoto’s thyroiditis. Material and methods: The study was conducted among 219 women aged 20–50 from southern Poland, using (i) Juczyński’s Healthy Behaviour Inventory (HBI); (ii) Beck’s Depression Inventory (BDI); (iii) satisfaction with life scale (SWLS) and (iv) WHOQoL-Bref (Quality of Life-BREFF). In the statistical analysis, Spearman’s R correlation coefficient and multiple regression analysis were applied, assuming a significance level of α = 0.05. Results: It was shown that with the increase in the general indicator of pro-health behaviours, the level of depressive symptoms decreased, while the level of satisfaction with life and all four aspects of quality of life on the WHOQoL scale increased (*p* < 0.001). Regression analysis demonstrated that the model consisting of all analysed pro-health behaviours explains a high percentage of variance in depressive symptoms (38%), life satisfaction (31%) and all aspects of quality of life, including those somatic and social (19%), psychological (28%) and environmental (12%). Conclusions: The noted correlations between pro-health behaviours, the intensity of depressive symptoms as well as the level of life satisfaction and quality of life indicate justification for promoting a pro-health lifestyle as a significant factor contributing to mental health and better quality of life among women with hypothyroidism.

## 1. Introduction

In the complex multifactorial aetiology of Hashimoto’s thyroiditis, in addition to genetic predispositions, modifiable environmental factors, including those related to lifestyle, play an important role. They significantly determine holistically defined health ([57]; [61]; [87]). In this context, an important element in the prevention and treatment of Hashimoto’s disease is a healthy lifestyle, including recreational physical activity, proper nutrition, effective strategies for coping with psychological stress, sleep and rest hygiene, avoiding tobacco smoke and other psychoactive substances, etc. Health-promoting behaviours also reduce the risk of cardio-metabolic diseases co-occurring with Hashimoto’s and improve overall health as well as quality of life ([13]; [15]; [19]; [38]; [53]; [63]; [69]; [71]; [81]; [93]; [96]).

Recreational physical activity and rational nutrition occupy a special place among health-promoting behaviours. Regular physical activity helps maintain proper body mass, body composition and physical fitness. It affects the normalisation of glycaemia and causes improvement in blood lipid profile. It also reduces psychological stress and improves overall mental state. Health benefits are brought by moderate-intensity physical activity of at least 150–300 min per week or vigorous-intensity physical activity of at least 75–150 min per week, supplemented by exercises that strengthen large muscle groups at least twice a week ([35]; [46]; [60]; [84]; [91]; [89]). The positive impact of physical activity on psychosocial health and quality of life has been confirmed in various population groups ([66]). A systematic review of the literature showed that training programs including aerobic endurance exercises had a positive effect on the physical and mental health of people with hypothyroidism ([20]). Additionally, a varied and balanced diet, taking the specificity of the disease and health condition into account, with the elimination of potential food antigens and processed foods, can reduce inflammation and the risk of hypothyroidism-related complications. The Mediterranean diet is particularly consistent with these recommendations ([13]; [14]; [15]; [19]; [38]; [39]; [61]; [63]; [69]; [71]; [81]; [88]; [93]). The Mediterranean diet, rich in vegetables, fruits, whole grains, olive oil, nuts and fish, can support thyroid function and reduce the risk of developing metabolic as well as mental disorders ([13]; [61]; [63]; [88]). In research, a significant positive correlation has been shown between compliance with recommendations of the Mediterranean diet and the levels of thyroxine, triiodothyronine and calcitonin in patients with subclinical hypothyroidism ([45]). A positive effect of a 10-week intervention program based on a normocaloric Mediterranean diet and increased physical activity was noted with regard to changes in anthropometric indicators, including reduction of adipose tissue, BMI and waist circumference. This positive effected was also demonstrated in women with polycystic ovary syndrome and Hashimoto’s disease ([46]). Despite the significant role of lifestyle in the prevention and treatment of hypothyroidism, poor knowledge on nutritional and dietary choices as well as a low level of physical activity have been reported among patients with Hashimoto’s thyroiditis, which may limit health potential ([1]; [11]; [28]; [45]; [61]; [94]).

According to the holistic model of health, pro-health behaviours, as components of lifestyle, significantly affect health potential, not only in the biological dimension, but also in the psychological and social ones. This is also important with regard to the risk of developing depression in the course of Hashimoto’s thyroiditis, in the etiopathogenesis of which an important role is played by disorders of the hypothalamic-pituitary-thyroid axis ([4]; [23]). Autoimmune thyroid diseases can lead to significant deterioration in quality of life dependent on health condition ([16]; [97]). Depressive disorders are diverse, and one of their types, according to the American Psychiatric Association (and the DSM-5 classification), comprises depressive disorders due to another medical condition ([37]). There are works available in the literature on the increased risk of depression in the course of Hashimoto’s thyroiditis ([4]; [54]; [58]; [78]; [97]). Depressive disorders are characterised by a low mood, anhedonia, increased fatigue and disturbances in appetite and sleep, as well as thoughts and actions of suicide. They are also often associated with a sense of persistent, chronic, undefined anxiety, which further impairs daily functioning ([23]; [78]; [97]).

A pro-health lifestyle, which helps to increase health potential, primary and secondary prevention and treatment of chronic diseases, is a factor related not only to health status, but also to quality of life, which is defined as an individual perception of one’s own life position, considering cultural conditions and one’s value system in connection with personal goals, expectations, norms and problems. Quality of life is influenced in a complex way through: physical health, interpersonal relations and environmental factors important for an individual. In the course of chronic disease treatment, including those autoimmune, in addition to achieving medical objectives, a significant role is played by the realisation of non-medical objectives, which include improving a patient’s well-being ([12]; [24]; [36]; [85]). Another element of quality of life is the level of satisfaction with life, which is a subjective indicator of well-being and is defined as general particularised cognitive assessment of life quality. Satisfaction with life directly refers to individual valuation of resources and limitations, finding expression in a positive attitude towards one’s own life situation. The sense of life satisfaction is one of the human health resources in the dimension of psychosocial health ([44]).

Chronic diseases affect quality of life. Research in this area has concerned various chronic diseases, including those thyroid-related ([76]; [86]) and others, i.e., diabetes, obesity, circulatory system, neurodegenerative and neurological diseases ([34]; [51]; [50]; [62]; [75]; [74]; [79]). In the area of life satisfaction, the literature includes works on the relationship between nutritional behaviours and the level of life satisfaction in menopausal women ([25], [26]), females with hypertension and type 2 diabetes ([27]; [29]) and physically active senior women from the Kraków population ([31]). A novel and unexploited area of research encompasses the issue of life quality behavioural determinants among people with autoimmune thyroid diseases, which are an important area of public health. In previous studies on this topic, assessment has been undertaken on the behavioural determinants of health ([28]; [56]; [73]) and quality of life among patients with various thyroid diseases ([76]; [86]). Our previous paper concerned the association between the level of physical activity, eating behaviours and depressive states in women with Hashimoto’s thyroiditis ([30]). However, there is a lack of more research gap concerning potential relationships between behavioural and psychological determinants of health and quality of life among people with thyroid diseases. Meanwhile, in order to increase the effectiveness of hypothyroidism treatment and its accompanying complications, a comprehensive diagnosis of the patient, taking mental state and quality of life into account, is crucial. The novelty of this research is its holistic dimension, taking behavioural and psychological aspects into account, including psychopathological ones.

The aim of the study was to gain knowledge about the relationship between health-promoting behaviours, depression and quality of life among women with Hashimoto’s thyroiditis. The following research questions were formulated: (a) What are the health-promoting behaviours, severity of depressive symptoms and level of life satisfaction and quality of life among women with Hashimoto’s? (b) What are the relationships between the level of pro-health behaviours and severity of depressive symptoms, as well as life satisfaction and quality of life among women with Hashimoto’s? 

The research hypothesis, assuming that health-promoting behaviours are associated with lower severity of depressive symptoms, higher level of life satisfaction and quality of life in women with Hashimoto’s disease, was empirically verified.

## 2. Materials and Methods

### 2.1. Participants

The study was conducted among women with Hashimoto’s thyroiditis from the southern region of Poland. The research was conducted by the authors in contact with patients of endocrinology clinics. A group comprising 219 women aged 20–50 was included in the study. The statistical assessment of the group size in the G*Power program showed that the required sample size was from 59 (for f2 = 0.35) to 129 individuals (for f2 = 0.15). Therefore, the study group met the criterion of the statistically expected number. The inclusion criteria for the study group were: female gender, being an adult, having a diagnosis of Hashimoto’s disease and providing written informed consent to participate in the study. Detailed characteristics of the group were described in our earlier paper ([30]) and are briefly presented in Table 1.

### 2.2. Research Tools

The study was conducted using standardised research tools in the field of health promotion and health psychology, including:(a)Juczyński’s Healthy Behaviour Inventory (HBI), which contains 24 statements describing various categories of health-related behaviours. The questionnaire was used to assess the general index of pro-health behaviours, as well as the results in four categories of healthy behaviours (positive mental attitude, preventive behaviours, proper eating habits and pro-health practices). The general index of pro-health behaviours (HBI Total) ranges from 24 to 120 points, with a higher score indicating a higher level of pro-health behaviours. The reliability of the tool was checked using Cronbach’s Alpha, which was 0.90 for the entire tool ([44]).(b)Beck’s Depression Inventory (BDI), which consists of 21 multiple choice questions (possible answers are scored from 0 to 3, i.e., from ‘no symptoms’ to a ‘strong symptom’). The questions refer to various symptoms of depression. In the interpretation of the results, it is assumed that obtaining 0–10 points means ‘no depression’ or ‘low depression’, i.e., ‘low mood’; 11–27 suggests ‘moderate depression’; and 28 or more indicates ‘severe depression’ ([6]; [43]). The severity of depressive symptoms in the studied group of women with Hashimoto’s thyroiditis was described in our previous paper. In the group was dominated by women without symptoms of depression or with low mood (53.9%), there were fewer women with moderate (38.4%) and severe depression (7.7%). The median scores in the BDI were 10.0, and mean scores were 11.9 ([30]).(c)The satisfaction with life scale (SWLS) in the Polish adaptation by Z. Juczyński is used to measure life satisfaction. It contains five statements. The examined person assesses to what extent each of them refers to their life so far. The result of the measurement is a general indicator of the sense of satisfaction with life. The reliability coefficient of Cronbach’s α = 0.81, and similarly, the scale stability index equalled 0.86. The results range from 5 to 35, and the higher the score, the higher the level of life satisfaction ([17]; [44]).(d)WHOQoL-bref (Quality of Life-BREFF) questionnaire, in the Polish adaptation by Wołowicka and Jaracz ([92]; [95]). The scale is used to assess life quality of healthy and ill individuals (for cognitive and clinical purposes). It contains 26 questions allowing to assess quality of life profile in the scope of four life dimensions: physical (somatic), psychological, social and environmental.

The research procedure is shown in Figure 1. The research protocol was approved by the Bioethics Committee at the District Medical Chamber in Kraków (No. 102/KBL/OIL/2019 dated 2 April 2019).

### 2.3. Statistical Analyses

Statistical calculations were performed in the Statistica 13.1 and JASP programs. The analysed variables were described using basic statistics (mean, median, standard deviation, minimum and maximum, lower and upper quartile). Due to the fact that the analysed variables were of non-normal distribution, the median was adopted as a measure of central tendency. In the statistical analysis (to determine the relationships between variables), correlation analysis was used, specifically Spearman’s non-parametric R (when at least one variable in the analysed pair demonstrated non-normal distribution). Multiple regression analysis was also implemented to determine the significance of models explaining the level of depressive symptoms, the level of life satisfaction and quality of life in the studied group of women. The significance level of α = 0.05 was adopted in the statistical analyses.

## 3. Results

### 3.1. Level of Pro-Health Behaviours Among Women with Hashimoto’s Disease

Among the domains of health-promoting behaviours (HBI), the surveyed women achieved the highest scores in the scope of preventive measures (Me = 23.0) and proper eating habits (Me = 22.0), followed by positive mental attitude (Me = 21.0) and pro-health practices (Me = 21.0). The overall index of health-promoting behaviours in the HB questionnaire was 87.0 (Table 2).

### 3.2. Level of Life Satisfaction and Quality of Life Among Women with Hashimoto’s Disease

In Table 3, results are presented regarding the level of satisfaction with life (SWLS) and the quality of life (WHOQoL-bref) among women. The median result for the satisfaction with life scale (SWLS), the total was 22.0. Among the areas of the quality of life scale (WHOQoL), women obtained raw results with the following medians: in the environmental (Me = 29.0), somatic and psychological domains (Me = 20.0), and in the social sphere (Me = 11.0). The calculated results (as the mean of the sum of items) for the WHOQoL scale are also presented, which allow to note that the highest medians concern the environmental area (Me = 14.5), while the lowest was found for the somatic sphere (Me = 11.43).

### 3.3. Health-Promoting Behaviours and Depressive Symptoms Among Women with Hashimoto’s Disease

The analyses indicated that all health-promoting behaviours and the general index of health-promoting behaviours were negatively associated with depressive symptoms, with the strongest association found with positive mental attitude and the general index of health-promoting behaviours (*p* < 0.001). This means that the higher the level of pro-health behaviours, the lower the level of depressive symptoms (Table 4).

Regression analysis (dependent variable—BDI, predictors—HBI) showed that the full model consisting of all analysed health behaviours (HBI) explains 38% of the variance with regard to depressive symptoms in the studied group of women (Table 5).

The predictors found to be significant were: positive mental attitude (negative) and pro-health practices (positive) (Table 6).

### 3.4. Health-Promoting Behaviours and Life Satisfaction Among Women with Hashimoto’s Disease

The analyses demonstrated that life satisfaction was positively associated with three domains of health-related behaviours, i.e., positive mental attitude (*p* < 0.001), preventive behaviours (*p* = 0.004) and pro-health practices (*p* < 0.001), as well as the general index of health-related behaviours (*p* < 0.001) (Table 7).

Regression analysis (dependent variable—SWLS, predictor individual dimensions of HBI) showed that the full model consisting of all analysed pro-health behaviour types was statistically significant and explained 31% of the variance concerning the level of life satisfaction in the study group (Table 8). Positive mental attitude turned out to be a significant positive predictor (Table 9).

### 3.5. Pro-Health Behaviours and Quality of Life Among Women with Hashimoto’s Disease

The analyses showed that positive psychological attitude was positively associated with all four aspects of quality of life (somatic, psychological, social and environmental) (*p* < 0.001). Similar relationships were described for preventive behaviours and all dimensions of the WHOQoL scale. Proper eating habits were positively correlated with the somatic (*p* < 0.001) and psychological aspects of life quality (*p* = 0.003). Also, another domain of pro-health behaviours—pro-health practices, was positively associated with the somatic (*p* = 0.006) and psychological (*p* = 0.022) aspects of life quality. At the same time, with the increase in the general index of health-promoting behaviours, the level of all four aspects of life quality on the WHOQoL scale experienced an increase (Table 10).

Regression analysis (dependent variables individual dimensions of WHOQoL, predictors of individual HBI dimensions) indicated that the full model consisting of all analysed pro-health behaviour types was statistically significant and explained 19% of the variance regarding the level of the somatic aspect of life quality, 28% of the psychological aspect of life quality, 19% of the social aspect and 12% of the environmental aspect of quality of life in the study group (Table 11). The only significant positive predictor in the case of the psychological, social and environmental aspects of life quality turned out to be positive mental attitude, while in the case of the somatic quality of life aspect, the positive, statistically significant predictors were proper eating habits and positive mental attitude (Table 12).

## 4. Discussion

In the presently discussed study, significant relationships were shown between pro-health behaviours and the intensity of depressive symptoms as well as satisfaction with life and quality of life among women with Hashimoto’s disease. This was done in such a way that allowed for positive verification of the adopted research hypothesis indicating that more intense pro-health behaviours are associated with a lower intensity of depressive symptoms and higher level of satisfaction with life as well as quality of life among women with Hashimoto’s thyroiditis. The regression analysis also confirmed that the model consisting of the analysed pro-health behaviours explains a high percentage of variance regarding depressive symptoms (38%), satisfaction with life (31%) and all aspects of quality of life, especially the psychological dimension (28%).

### 4.1. Pro-Health Behaviours, Depressive Symptoms, Satisfaction with Life and Quality of Life Among Women with Hashimoto’s Disease

An important aspect of therapeutic treatment in Hashimoto’s disease, apart from the therapy of hypothyroidism treated by taking thyroid hormones (levorotatory thyroxine), is a healthy lifestyle, including a wide range of health-promoting behaviours ([82]). In the discussed research, an average level of health-promoting behaviours in the lifestyle was demonstrated for women with Hashimoto’s disease (HBI Total at the level of 87 points, i.e., 6th sten according to the provisional Polish norms) ([44]). Among the analysed areas of health-promoting behaviours, women paid the greatest attention to preventive (adherence to medical recommendations, regular medical examinations) and proper eating habits, followed by positive mental attitude (positive thinking, maintaining proper relationships with other people, avoiding strong emotions and tension) and pro-health practices (adequate amount of sleep and rest, undertaking recreational physical activity, limiting stimulants). The highest level of preventive behaviours indicates that women attach great importance to medical examinations preventing the deterioration of health. The obtained results correspond to those obtained in research carried out by other authors ([77]). Thus, in other studies conducted among individuals with hypothyroidism, an average intensity of health-related behaviours was also exhibited (approximately 83 out of 118 max), thus similarly to the current study, whereby the intensity of health-promoting behaviours increased, among others, along with the increase in disease acceptance (especially in terms of positive mental attitude) ([77]). Furthermore, in another group of women with Hashimoto’s disease, an average level of pro-health behaviours was found (approximately 80 points), as well as the dominance of women with a low level of HBI Total (approximately 41%) ([42]). Similar trends, i.e., an average level of health-related behaviours and a high level of those preventive, with a low level of other pro-health behaviour domains, were described among patients experiencing heart failure, with women generally obtaining higher results than men ([49]). Other trends were demonstrated in studies on young and healthy people, including among academic youth, for instance, among students of various fields from the Kraków environment, who obtained the highest results in the area of positive mental attitude and the lowest in the category of preventive behaviours ([48]).

Various levels of depression symptoms were found in the studied group of women; although women without symptoms of depression predominated (approximately 54%), approximately 38% showed moderate depression, and nearly 8% severe depression ([30]). In other studies carried out at various research centres, the significant prevalence was indicated of mood disorders, anxiety and depression among individuals with Hashimoto’s thyroiditis ([4]; [7]; [54]; [58]; [82]; [97]). The meta-analysis also confirmed a two to three times higher risk of developing anxiety and depressive disorders in patients with Hashimoto’s thyroiditis compared to the control group ([78]).

Satisfaction with life assessed in the present study, as a personal resource related to health culture, is understood as a subjective indicator of life quality, expressed in the sense of satisfaction with one’s own achievements and living conditions ([44]). In research, it has been confirmed that with an increase in sense of satisfaction, levels of anger, anxiety and depression decrease ([16]). Among the studied women with Hashimoto’s disease, an average level of life satisfaction was demonstrated (median of 22, which is 6th sten according to the provisional Polish norms) ([44]). In other studies among women with type 2 diabetes and those at a perimenopausal age, a similar level of life satisfaction was demonstrated, measured with the SWLS ([27]; [29]). It can therefore be stated that the level of life satisfaction among the studied women with Hashimoto’s disease does not significantly differ from the values described in other groups of women, both healthy and with chronic diseases.

For quality of life, assessed using the WHOQoL-bref scale, it was shown that women with Hashimoto’s thyroiditis achieved the highest results in the social and environmental domains and the lowest in the somatic sphere. Low values of the somatic domain may be explained by the existing disease state as well as related somatic symptoms and complications. Assessment of health-associated quality of life (HRQoL) among people with hypothyroidism was also the subject of research conducted by other authors. Thus, in studies among patients, including women with hypothyroidism, it was demonstrated that individuals with hypothyroidism achieved the lowest values in the environmental domain and the highest in the social sphere. At the same time, it was shown that both women and men with hypothyroidism achieved lower values in three domains of quality of life (somatic, psychological and social) compared to the control group ([72]). The mean values of all life quality domains on the WHOQoL scale were shown to increase in other studies related to this area and carried out among patients in pre- and post-operative periods ([18]). In other studies, women with Hashimoto’s disease scored lower in all domains of quality of life (including mental health) than the control group ([97]). An average level of life quality was also described in people with heart failure ([49]). In another group of people with type 2 and 1 diabetes, the highest life satisfaction was demonstrated in the social domain, followed by physical and environmental, and the lowest in the psychological sphere. There was also differentiation in the level of life satisfaction depending on health status (without complications vs. with complications), the greatest differences being present in the physical and psychological domains ([51]). Low quality of life due to the disease was also noted in patients with hypothyroidism ([64]; [67]).

### 4.2. Correlations Between Pro-Health Behaviours, Depressive Disorders, Satisfaction with Life and Quality of Life Among Women with Hashimoto’s Disease

In the discussed research, it was shown that all domains of pro-health behaviours and the general index of health-promoting behaviours were negatively associated with depressive symptoms, with the strongest correlation observed between positive mental attitude and the general index of pro-health behaviours. Therefore, the lower the level of pro-health behaviours, the higher the intensity of depressive symptoms (and vice versa). The performed multiple regression analysis additionally confirmed that the model consisting of the analysed health-promoting behaviours explains a high percentage of variance concerning depressive symptoms in women (38%). Thus, the results allowed for positive verification of the research hypothesis adopted in this area. The obtained results confirmed that pro-health behaviours, including positive mental attitude, are of predictive significance for depressive states in women with Hashimoto’s. At the same time, they correspond to the results of studies by other authors, in which a relationship is indicated between health-promoting behaviours (so-called lifestyle medicine), including physical activity, a rational diet, sleep and rest hygiene, good social interactions, coping with psychological stress and not using psychoactive substances, as well as a lower risk of depression and more effective treatment ([75]; [41]). The importance of an active lifestyle (physical exercise) in alleviating depressive symptoms has also been confirmed another study ([52]). Our research among women with Hashimoto’s disease also showed, that women with low levels of depressive symptoms demonstrated higher levels of vigorous physical activity than women with moderate and high levels of depression ([30]). In various studies, the importance of lifestyle interventions has been supported, including increased physical activity and integrated models of depression treatment ([55]; [59]; [70]). The predictive significance of pro-health behaviours for the risk of depression has also been described during the COVID-19 pandemic ([10]). In other studies, the importance has been confirmed of nutrition and diet quality for mental health ([3]; [9]; [13]; [21]; [22]; [33]; [68]; [83]; [90]; [98]; [99]). For example, in the European PREDIMED study, not only the cardioprotective significance of the Mediterranean diet was verified, but also its role in reducing the risk of developing depression ([8]). Moreover, in the Australian Smiles study, it was shown that people with depression who underwent a three-month dietary intervention (Mediterranean diet) achieved a significantly greater reduction in depressive symptoms than the control group (without dietary intervention, but with pharmacotherapy or psychotherapy) ([40]). In another Australian study—HELFIMED—it was demonstrated that the use of the Mediterranean diet (more vegetables, fruits, nuts, whole grains and legumes, and less red meat and poultry) was connected with a reduction in depressive symptoms and improved quality of life ([65]). Our research also showed significant relationships between dietary behaviours and depressive symptoms, with some rational dietary choices (including consumption of products rich in zinc) being associated with a lower intensity of depressive symptoms in women with Hashimoto’s disease ([30]). A systematic review of the literature also confirmed that high consumption of fruits, especially berries and citrus fruits, as well as vegetables, especially green leafy vegetables, contributed to increased optimism and reduced stress and depression ([33]). In turn, frequent consumption of fast food and fried food (and acrylamide) increased the risk of depression and anxiety due to disorders of lipid metabolism in the brain and inflammation in the nervous system ([83]; [90]). The risk of depression was also increased by poor health quality of the diet, including low supply of omega-3 unsaturated acids, vitamins (B6, B9, B12, D), minerals (magnesium, selenium, zinc) and prebiotics as well as probiotics, and excess cholesterol and sodium in the diet ([21]; [98]).

In terms of the relationships between health-promoting behaviours as well as satisfaction with and quality of life among women with Hashimoto’s, in the present trial, it was noted that individual domains of pro-health behaviours, including positive mental attitude and preventive behaviours, as well as the general indicator of health-promoting behaviours, were positively associated with life satisfaction and all four aspects of quality of life (somatic, psychological, social and environmental). In turn, other domains of healthy behaviours, i.e., proper eating habits and pro-health practices (sleep and rest hygiene, physical activity, limiting stimulants), were positively associated with the somatic and psychological aspects of quality of life. The performed multiple regression analysis additionally confirmed that the model consisting of the analysed pro-health behaviours explains a high percentage of variance in life satisfaction (31%) and all aspects of quality of life in women, including those psychological (28%), somatic and social (18%) and environmental (12%), with a special predictive role of positive mental attitude and proper eating habits. Therefore, the obtained results positively verified the research hypothesis adopted in this area. In other studies, it has also been shown that the life quality of cardiology patients increased with the intensification of health-promoting behaviours ([49]). In other studies conducted in Silesia (Poland) among a group of young women with Hashimoto’s disease, it was demonstrated that improvement in quality of life and increased life satisfaction were facilitated by a higher level of support ([5]). In another population group comprising Kraków university students, a positive relationship was found between the intensity of individual health-promoting behaviour categories and the level of life satisfaction, with the strongest relationship noted between sense of life satisfaction and positive mental attitude ([47]), which corresponds to the results discussed among women with Hashimoto’s. Also, in research carried out among middle-aged women of Arab origin in the USA, it was indicated that health-promoting behaviours were significantly correlated with personal, health and life satisfaction ([2]). Physical activity in one’s lifestyle, as one of the pro-health behaviours, was associated with greater life satisfaction, also in young adults in Poland ([80]). The relationships between personal resources, including life satisfaction and dietary choices, have also been described in perimenopausal women, females with type 2 diabetes and elderly women from the Kraków population (Poland), with an indication of more rational ones among women who are more satisfied with their lives ([25], [26], [27]; [29]; [31]). In elderly women, life satisfaction was shown to be positively associated with regular meal consumption, proper fluid replenishment, daily consumption of whole grain products, vitamin D supplementation, avoiding alcoholic beverages and limiting red meat as well as animal fats in the diet ([31]). A systematic review of the literature also confirmed that quality of life, well-being and psycho-emotional balance were significantly associated with diet quality, physical activity level, sleep and rest hygiene, and avoidance of psychoactive substances ([32]).

Concluding, the discussed research conducted among women with Hashimoto’s in this study, as well as in research by other authors, confirmed significant positive correlations between pro-health behaviours, i.e., a pro-health lifestyle, including, in particular, positive mental attitude and proper dietary habits, with a higher quality of life and life satisfaction, which is a cognitive dimension of life quality.

### 4.3. Limitations and Directions for Further Research

The limitations of the work are related to, among others, the self-reported nature of the research tools used. The cognitive value of the work would also be increased by including a control group in the study, as well as by assessing the predictive role of socio-demographic characteristics. The results were obtained among women mostly from large urban centres with higher education, covered by medical care; thus, they cannot be representative of all women with Hashimoto’s. The cross-sectional nature of the study is also a limitation. The indicated limitations restrict the possibility of generalising the results. As part of continuation of research in this area, it would also be interesting to assess the relationships between other aspects of lifestyle, including the way of coping with psychological stress and conducting multifaceted assessment regarding nutrition habits and diet health quality with depressive symptoms and quality of life among women with Hashimoto’s. In addition, it would be possible to analyse selected health indicators (including body mass and composition, as well as morphology and biochemistry results of peripheral blood and urine) in the context of depressive symptoms and quality of life among patients with hypothyroidism. The research could be extended beyond women to other groups of patients, and other variables could be taken into account, including socio-demographic characteristics. The strength of the study seems to be the multifaceted nature of the research, in the field of health promotion and health psychology.

## 5. Conclusions

Among women with Hashimoto’s disease, an average level of pro-health behaviours, varied level of depressive symptoms, average levels of life satisfaction and differentiation of individual domains of life quality were found, with the lowest level of the somatic domain indicated.

Significant correlations were found between pro-health behaviours and the intensity of depressive symptoms as well as satisfaction with life and quality of life among women with Hashimoto’s disease. The more intense pro-health behaviours were associated with a lower intensity of depressive symptoms and higher levels of satisfaction with life and quality of life among women with Hashimoto’s thyroiditis.

In light of the obtained results, it seems reasonable to promote a pro-health lifestyle, with a wide spectrum of pro-health behaviours, as an important factor that promotes not only holistically understood health, but also a better quality of life among women with hypothyroidism, which should be applied in practice.

## Figures and Tables

**Figure 1 ejihpe-15-00097-f001:**
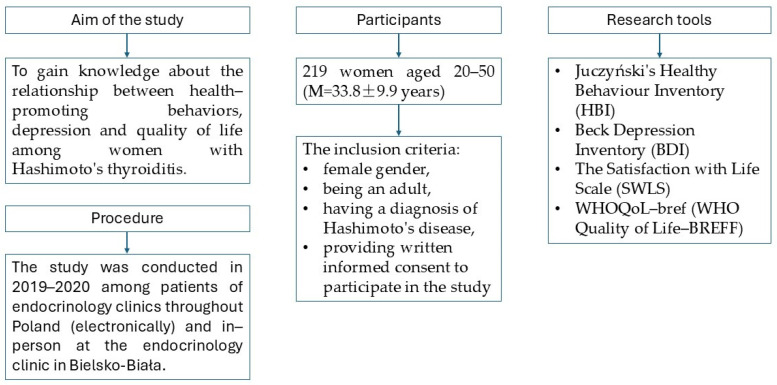
Research procedure diagram.

**Table 1 ejihpe-15-00097-t001:** Sociodemographic characteristics and BMI of women with Hashimoto’s disease.

Variables		%
Place of residence	City with above 100,000 residents	40.4
	City with below 100,000 residents	35.8
	Villages	23.8
Level of education	Higher	77.9
	Secondary	19.3
	Vocational	2.8
Body Mass Index	Normal weight BMI 18.5–24.9 kg/m^2^	52.1
	Overweight BMI 25.0–29.9 kg/m^2^	29.7
	Obesity BMI ≥ 30 kg/m^2^	12.8
	Underweight BMI < 18.5 kg/m^2^	5.4

**Table 2 ejihpe-15-00097-t002:** Level of health-promoting behaviours (HBI) among women with Hashimoto’s disease (*n* = 219, descriptive statistics).

Variables		M	Me	Min	Max	Q25	Q75	SD
HBI	Positive mental attitude	21.21	21.00	10	29	19.00	24.00	3.75
	Preventative behaviours	22.22	23.00	11	30	20.00	25.00	3.88
	Proper eating habits	21.62	22.00	9	30	19.00	25.00	4.58
	Pro-health practices	20.80	21.00	6	30	18.00	24.00	3.89
	HBI Total	85.85	87.00	53	112	78.00	93.00	11.30

M—mean, Me—median, Min—minimum, Max—maximum, Q25—lower quartile, Q75—upper quartile, SD—standard deviation.

**Table 3 ejihpe-15-00097-t003:** Level of life satisfaction (SWLS) and quality of life (WHOQoL) among women with Hashimoto’s disease (*n* = 219, descriptive statistics).

Variables		M	Me	Min	Max	Q25	Q75	SD
SWLS		20.82	22.00	5	35	17.00	25.00	5.66
WHO-QoL	Raw results							
	WHOQOL Somatic	20.24	20.00	11	30	18.00	23.00	3.26
	WHOQoL Psychological	20.15	20.00	9	27	18.00	22.00	3.18
	WHOQoL Social	10.71	11.00	3	15	9.00	13.00	2.66
	WHOQoL Environmental	28.37	29.00	12	39	26.00	31.00	4.65
	Converted results (scale in accordance with WHOQoL-100)							
	WHOQoL Somatic	11.57	11.43	6.29	17.14	10.29	13.14	1.86
	WHOQoL Psychological	13.43	13.33	6.00	18.00	12.00	14.67	2.12
	WHOQoL Social	14.28	14.67	4.00	20.00	12.00	17.33	3.54
	WHOQoL Environmental	14.19	14.50	6.00	19.50	13.00	15.50	2.33

M—mean, Me—median, Min—minimum, Max—maximum, Q25—lower quartile, Q75—upper quartile, SD—standard deviation.

**Table 4 ejihpe-15-00097-t004:** Relationships between health-promoting behaviours (HBI) and severity of depressive symptoms (BDI) among women with Hashimoto’s disease (*n* = 219, Spearman’s R correlations).

Variables	Spearman’s R	t(*n* − 2)	*p*
Positive mental attitude and BDI	−0.57	−10.24	<0.001
Preventative behaviours and BDI	−0.20	−3.07	0.002
Proper eating habits and BDI	−0.20	−3.07	0.002
Pro-health practices and BDI	−0.16	−2.32	0.021
HBI Total and BDI	−0.38	−6.06	<0.001

R—Rho, t—Student’s *t*-test, *p*—*p*-value.

**Table 5 ejihpe-15-00097-t005:** Regression analysis—BDI and HBI (full model).

	Multiple—R	Multiple—R^2^	Adjusted—R^2^	df—Model	df—Residual	F	*p*
BDI	0.62	0.38	0.37	4	214	33.2	<0.001

R—correlation coefficient, R^2^—coefficient of determination, df—degrees of freedom, F—F-statistic, *p*—*p*-value.

**Table 6 ejihpe-15-00097-t006:** Regression coefficients—BDI and HBI (full model).

	BDI—Param.	BDI—Std.Err	BDI—t	BDI—*p*	BDI—Beta (ß)	BDI—St.Err.ß
Intercept	43.06	3.997	10.8	<0.001		
Positive mental attitude	−1.63	0.154	−10.6	<0.001	−0.64	0.06
Preventative behaviours	−0.00	0.157	−0.0	0.986	−0.00	0.06
Proper eating habits	−0.14	0.132	−1.1	0.279	−0.07	0.06
Pro-health practices	0.31	0.147	2.1	0.035	0.13	0.06

Param.—regression coefficient, Std.Err—standard error, t—Student’s *t*-test, Beta—standardized regression coefficient, *p*—*p*-value.

**Table 7 ejihpe-15-00097-t007:** Correlations between health-promoting behaviours (HBI) and level of life satisfaction (SWLS) among women with Hashimoto’s disease (*n* = 219, Spearman’s R correlations).

Variables	Spearman’s R	t(*n* − 2)	*p*
Positive mental attitude and SWLS	0.49	8.30	<0.001
Preventative behaviours and SWLS	0.19	2.89	0.004
Proper eating habits and SWLS	0.11	1.62	0.107
Pro-health practices and SWLS	0.26	3.94	<0.001
HBI Total and SWLS	0.34	5.27	<0.001

R—Rho, t—Student’s *t*-test, *p*—*p*-value.

**Table 8 ejihpe-15-00097-t008:** Regression analysis—SWLS and HBI (individual dimensions).

	Multiple—R	Multiple—R^2^	Adjusted—R^2^	df—Model	df—Residual	F	*p*
SWLS	0.56	0.31	0.30	4	214	24.41	<0.001

R—correlation coefficient, R^2^—coefficient of determination, df—degrees of freedom, F—F-statistic, *p*—*p*-value.

**Table 9 ejihpe-15-00097-t009:** Regression coefficients—SWLS and HBI (individual dimensions).

	SWLS Beta (ß)	SWLS St.Err.ß	SWLS Param.	SWLS Std.Err	SWLS t	SWLS *p*
Intercept			1.83	2.50	0.73	0.466
Positive mental attitude	0.50	0.06	0.76	0.10	7.85	<0.001
Preventative behaviours	0.08	0.07	0.11	0.10	1.14	0.256
Proper eating habits	−0.11	0.07	−0.13	0.08	−1.61	0.109
Pro-health practices	0.11	0.06	0.16	0.09	1.76	0.080

Param.—regression coefficient, Std.Err—standard error, t—Student’s *t*-test, Beta—standardized regression coefficient, *p*—*p*-value.

**Table 10 ejihpe-15-00097-t010:** Correlations between health-promoting behaviours (individual domains and the general HBI index) and quality of life (WHOQoL) among women with Hashimoto’s disease (*n* = 219, Spearman’s R correlations).

Variables	Spearman’s R	t(*n* − 2)	*p*
Positive mental attitude and WHOQoL Somatic	0.32	4.99	<0.001
Positive mental attitude and WHOQoL Psychological	0.46	7.66	<0.001
Positive mental attitude and WHOQoL Social	0.37	5.78	<0.001
Positive mental attitude and WHOQoL Environmental	0.29	4.43	<0.001
Preventative behaviours and WHOQoL Somatic	0.20	2.98	0.003
Preventative behaviours and WHOQoL Psychological	0.24	3.61	<0.001
Preventative behaviours and WHOQoL Social	0.19	2.87	0.005
Preventative behaviours and WHOQ_ Environmental	0.16	2.35	0.02
Proper eating habits and WHOQoL Somatic	0.38	5.99	<0.001
Proper eating habits and WHOQoL Psychological	0.20	3.00	0.003
Proper eating habits and WHOQoL Social	0.06	0.87	0.387
Proper eating habits and WHOQoL Environmental	0.10	1.45	0.148
Pro-health practices and WHOQoL Somatic	0.18	2.76	0.006
Pro-health practices and WHOQoL Psychological	0.15	2.31	0.022
Pro-health practices and WHOQoL Social	0.05	0.76	0.451
Pro-health practices and WHOQoL Environmental	0.12	1.85	0.066
HBI Total and WHOQoL Somatic	0.39	6.32	<0.001
HBI Total and WHOQoL Psychological	0.37	5.88	<0.001
HBI Total and WHOQoL Social	0.22	3.28	0.001
HBI Total and WHOQoL Environmental	0.22	3.36	0.001

R—Rho, t—Student’s *t*-test, *p*—*p*-value.

**Table 11 ejihpe-15-00097-t011:** Regression analysis—WHOQoL and HBI (individual dimensions).

WHOQoL	Multiple—R	Multiple—R^2^	Adjusted—R^2^	df—Model	df—Residual	F	*p*
Somatic	0.44	0.19	0.18	4	214	12.74	<0.001
Psychological	0.53	0.28	0.27	4	214	21.00	<0.001
Social	0.44	0.19	0.18	4	214	12.76	<0.001
Environmental	0.34	0.12	0.10	4	214	7.13	<0.001

R—correlation coefficient, R^2^—coefficient of determination, df—degrees of freedom, F—F-statistic, *p*—*p*-value.

**Table 12 ejihpe-15-00097-t012:** Regression coefficient—WHOQoL and HBI (individual dimensions).

WHOQoL Somatic						
	Beta (ß)	St.Err.ß	Param.	Std.Err	t	*p*
Intercept			11.07	1.56	7.08	<0.001
Positive mental attitude	0.28	0.07	0.24	0.06	4.02	<0.001
Preventative behaviours	−0.05	0.07	−0.04	0.06	−0.66	0.507
Proper eating habits	0.31	0.07	0.22	0.05	4.27	<0.001
Pro-health practices	0.01	0.07	0.01	0.06	0.17	0.861
**WHOQoL Psychological**						
	**Beta (ß)**	**St.Err.ß**	**Param.**	**Std.Err**	**t**	** *p* **
Intercept			9.74	1.43	6.79	<0.001
Positive mental attitude	0.50	0.07	0.42	0.06	7.62	<0.001
Preventative behaviours	0.08	0.07	0.07	0.06	1.17	0.244
Proper eating habits	−0.01	0.07	−0.01	0.05	−0.14	0.889
Pro-health practices	0.01	0.06	0.01	0.05	0.15	0.882
**WHOQoL Social**						
	**Beta (ß)**	**St.Err.ß**	**Param.**	**Std.Err**	**t**	** *p* **
Intercept			4.50	1.27	3.53	0.001
Positive mental attitude	0.42	0.07	0.30	0.05	6.14	<0.001
Preventative behaviours	0.12	0.07	0.08	0.05	1.61	0.108
Proper eating habits	−0.06	0.07	−0.04	0.04	−0.85	0.397
Pro-health practices	−0.08	0.07	−0.06	0.05	−1.23	0.220
**WHOQoL Environmental**						
	**Beta (ß)**	**St.Err.ß**	**Param.**	**Std.Err**	**t**	** *p* **
Intercept			19.29	2.33	8.28	<0.001
Positive mental attitude	0.34	0.07	0.42	0.09	4.69	<0.001
Preventative behaviours	0.04	0.08	0.05	0.09	0.54	0.592
Proper eating habits	−0.01	0.08	−0.01	0.08	−0.13	0.897
Pro-health practices	−0.03	0.07	−0.03	0.09	−0.40	0.687

Param.—regression coefficient, Std.Err—standard error, t—Student’s *t*-test, Beta—standardized regression coefficient, *p*—*p*-value.

## Data Availability

Data are available on request from the authors.

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
