# Peer review of "Pro-Health Behaviours and Depressive Symptoms as Well as Satisfaction with and Quality of Life Among Women with Hashimoto’s Disease"

_ejihpe, 2025, doi:10.3390/ejihpe15060097_

Round 1
Reviewer 1 Report
Comments and Suggestions for Authors
Dear Authors,
the aim of the study was to analyze the relationships between health-promoting behaviors, depression, and quality of life among women with Hashimoto's thyroiditis - rather - the aim of the study was to gain knowledge about the relationships between health-promoting behaviors, depression, and quality of life among women with Hashimoto's thyroiditis. You went on to write that lifestyle is one of the important factors determining health and quality of life. You point to recreational physical activity and a balanced diet, and then to health training, which is a broader concept—rather—forms of physical activity and a tailored diet and properly organized health training. These premises need to be clarified. See and consider: doi: 10.3389/fpsyg.2024.1537842 ; doi: 10.3390/ijms25115874 ; doi: 10.1002/ijgo.70156). The assumptions and objectives have been correctly formulated. I have no reservations about the methodology – it is clear and unambiguous. However, explanations of the symbols used under the tables should be added. The discussion is divided into logical subsections, but not sufficiently. The following subsections should be added: “Directions for further research” and “Practical recommendations,” which may contribute to the dynamic implementation of your research results.
Thank you!
Reviewer 2 Report
Comments and Suggestions for Authors
Dear Authors,
Thank you for sending me your work for review,
you are taking up an important research topic, especially since more and more people are suffering from Hashimoto's.
The work requires additions and corrections within individual sections:
Introduction:
-in the introduction, please expand on the topic of the influence of specific components of lifestyle in people suffering from Hashimoto's. You do not provide any research results on the impact of specific forms of physical activity or nutritional interventions on sick people, which you must supplement by citing specific literature
Material and methods:
- in this section, please add a graph illustrating the course of the research experiment, taking into account the number, duration and criteria for inclusion and exclusion from the study
- please add a table of the characteristics of the research group, the information contained in the article that a detailed description of the group is in another work is insufficient
Results:
In the tables, please highlight statistically significant results
- please present key results in the form of graphs
Discussion:
please add in the discussion research results of other authors showing how physical activity or a specific way of eating (lifestyle) can affect the quality and satisfaction of life, reduce stress or depression
- please add a section of limitations, and describe in it what occurred during the implementation of this project. Please also indicate the strengths and weaknesses conducted research
best regards
Round 2
Reviewer 2 Report
Comments and Suggestions for Authors
Dear authors, thank you for following my guidelines and suggestions,
in the material and methods section please add a graph illustrating the course of the research experiment, which will be legible for potential recipients,
with respect
Author Response
Dear Reviewer,
I have completed the recommended graph with pleasure, thank you very much for your kindness, with respect, MG